# Interspecific Hybridization between the Two Sympatric Termite *Reticulitermes* Species under Laboratory Conditions

**DOI:** 10.3390/insects11010014

**Published:** 2019-12-23

**Authors:** Jia Wu, Huan Xu, Ali Hassan, Qiuying Huang

**Affiliations:** Hubei Insect Resources Utilization and Sustainable Pest Management Key Laboratory, Huazhong Agricultural University, Wuhan 430070, China; wujia831019@163.com (J.W.); huanxu@webmail.hzau.edu.cn (H.X.); alihassan213d@yahoo.com (A.H.)

**Keywords:** termites, hybridization, mating behavior, partners, genotypes

## Abstract

Closely related species have similar reproductive behaviors and recognition systems, which contribute to interspecific interactions. However, few studies have explored interspecific reproduction choice and mating in termites. We investigated whether hybridization between two sympatric termites, *Reticulitermes flaviceps* and *R. chinensis*, occurs under laboratory conditions. We found that frequencies of acceptance were significantly higher than those of agonism between interspecific partners. There were no significant differences in frequencies of tandem and mating behaviors between intraspecific and interspecific partners. However, the allogrooming frequencies of interspecific partners were significantly higher than intraspecific partners. There were no significant differences in the duration of tandem, allogrooming, or mating behavior at each time between conspecific partners and heterospecfic partners. Genotyping analyses further showed that both intraspecific and interspecific mating were able to produce offspring. We conclude that interspecific hybridization does occur between two termite *Reticulitermes species* under laboratory conditions.

## 1. Introduction

Hybridization between relative species is considered to be a potential way to increase the chance of mating and reproduction for individuals that fail to mate with intraspecific partners. Closely related species often have similar reproductive behaviors and recognition systems, which contribute to interspecific interactions. However, behavioral preference for conspecifics may form some barrier limiting interaction and results in minimum chances of gene flow between species [1,2,3]. Behavioral observations have suggested that preferences for conspecific may be plastic [1,4]. It can be minimized or reversed depending on the fitness of offspring and environments [1,2,3]. While the preference for conspecifics plays a role in keeping a pure gene pool in each species, its breakdown may facilitate hybridization between species [4].

In previously described cases of interspecies mating in termites, the two productive individuals from different species were forced to pair, and they had no chance to choose their partners between interspecifics and intraspecifics [5,6]. For example, interspecific mating occurred between *Nasutitermes corniger* Motschulsky and *Nasutitermes ephratae* Holmgren in lab [6]. Similarly, interspecific mating was found in invasive species *Coptotermes formosanus* Shirak and *Coptotermes gestroi* Wasmann in south Florida [7]. The above cases indicate that hybridization between two relative termite species were able to produce live offspring. In termites, the sex-pairing pheromone is similar in sibling species [5,6,8], and neither queens nor kings possess external sclerotized genitalia, which results in no physical barriers to interspecies mating [9]. The indications of preferences for conspecifics may be largely absent in many termite sibling species. If so, it may be possible to create a termite colony between interspecies.

*Reticulitermes flaviceps* Oshima and *Reticuliterme chinensis* Snyder are consistently recognized as the two distinct species on the basis of multiple criteria including morphology, biology and mitochondrial genomic characteristics [10,11]. *R. flaviceps* and *R. chinensis* share habitat, including nesting and foraging sites. Despite asynchronous peak flights, there is some overlap in their dispersal flight seasons. Therefore, we speculate that reproductive individuals of the two termite species may hybridize when they encounter each other in nature.

To demonstrate our speculation, frequencies and duration of several behaviors, which were acceptance, agonism, tandem, allogrooming interspecies, and intraspecies mating, were measured and compared in *R. flaviceps* and *R. chinensis*. We also analyzed genotypes of parents and offspring from five colonies built by interspecific partners and intraspecific partners using five microsatellite loci. Our results will provide new insights into reproductive barriers and hybridization between the sympatric termite species.

## 2. Materials and Methods

### 2.1. Termites

All *R. chinensis* and *R. flaviceps* colonies used in this study were collected from Shizi hill, Wuhan city, Hubei province, China from March to April (swarming seasons) of 2018 (Appendix A). Images of *R. chinensis* and *R. flaviceps* differ in having black or yellow pronota, respectively. Alates living in dead logs were brought to the laboratory and kept under moisture 75% at a temperature of 25 °C to promote dispersal flight. Subsequently, individual alates were removed from dead logs and their genders were identified via the shape of the seventh abdominal sternite [12,13]. Individuals with the same sex from the same colony in each species were placed in Petri-dishes with a small piece of wood and a moistened filter paper for pairing experiments commencing of the day when nestmates in their home colony disperse.

### 2.2. Experimental Setup

Five *R. flaviceps* colonies were labeled A_1_ to A_5,_ and five *R. chinensis* colonies were labeled B_1_ to B_5_. We set five groups: A_1_B_1_, A_2_B_2_, A_3_B_3_, A_4_B_4_ and A_5_B_5_. Each group was composed of a *R. flaviceps* couple and a *R. chinensis* couple. Male individuals for both species were marked with white color (uni-Paint markers PX-21, Mitsubishi Pencil Company, Tokyo, Japan) on their abdomens. Each group was placed into 60 mm Petri-dishes with moistened filter paper. We established the following experiment: (1) the behavioral observation of agonism, acceptance, tandem and allogrooming. Videos (3 min long) were taken with a high definition (HD) camera (Nikon D7000 with 60 mm lens, Tokyo, Japan) at four times after the groups were established, at 10 min, 30 min, 50 min and 70 min, because the identification and tandem behavior of dealates was duration half hour to one hour after encountering each other. Six replicates for each group were used. (2) After above behavior finished, four group including A_1_B_1_, A_2_B_2_, A_3_B_3_, and A_4_B_4_ were used in the behavioral observation of mating. One-hour videos were taken by every two hours. We did 2 replicates for each group and thus there were 8 replicates. (3) After the filming was finished, the dealates were placed into a 120 mL transparent cylindrical vial (ϕ = 3 cm) with moistened filter paper and pine wood at 20–26 °C in constant darkness.

### 2.3. Behavioral Observation

The reproductive behavior of termites from swarming to mating has been described in previous studies [9,14]. In this study we viewed the videos to determine the following behaviors. (1) Agonism: a quick attack with open mandibles towards an alien with intension to bite followed by return to original state. (2) Acceptance: if there is no aggression between encountering individuals, we assume that they accept each other. (3) Tandem: the male follows the female and antennates the posterior part of the female abdomen when searching for an adequate nesting site. (4) Allogrooming: we define a cleaning of each other’s bodies as a nuptial gift for mating. (5) Mating: Termites copulate in the opposite position by joining their abdominal ends.

If the events of tandem/allogrooming/mating behavior continued more than 5 s, the behavior was considered valid. If the interval time was less than 3 s between two events of tandem/mating, the behavior was recorded as one event. General Linear Mixed Models (GLMM) were used to statistically analyze behavioral frequency difference between interspecies and intraspecies, and performed using SPSS v19 (IBM Corp., Armonk, NY, USA). For mixed-effects models, the groups were considered to be random effects and the type of behaviors were considered to be a fixed effect. We applied an independent sample t-test to analyze the duration of behaviors between interspecies and intraspecies. All values were expressed as the mean ± SEM. The *p* value less than 0.05 was considered to be statistically significant.

### 2.4. Genotyping Analyses

To evaluate hybridization or conspecific mating between *R. chinensis* and *R. flaviceps*, we analyzed genotypes of the parents and their offspring from the five colonies built by interspecific partners and intraspecific partners using five microsatellite loci. The colonies were assayed three months after post-colony-establishment. DNA was extracted from each individual (detailed sample data see Appendix A) using a TIANamp Genomic DNA Kit (Tian Gen biotech Co., Ltd., Beijing, China) according to the recommendations of the manufacturers. Each DNA sample was amplified by PCR and performed using five pairs of microsatellite primers: Rs03, Rs78, Rs76, Ra144 and Ra141 (Appendix A) and the conditions for each PCR have been improved base on the research conducted by Wu et al. [13]. A color marker was added to 5′-primers and all primers synthesized by Invitrogen Trading (Shanghai, China) Co., Ltd. GeneMapper v4.0 (Applied Biosystems, Foster City, CA, USA) and Peak Scanner v1.0 (Applied Biosystems, Foster City, CA, USA) were used for analysis. Both the *R. flaviceps* and *R. chinensis* are non-parthenogenetic species [15]. The offspring produced by sexual reproduction were diploid. Thus, we estimated the offspring produced by hybridization or conspecific mating via microsatellite loci.

## 3. Results

Agonism reduced mating frequency, while acceptance represents more possibility of mating when interspecific reproductives encounter each other. Our results indicated that the frequencies of acceptance were significantly higher than those of aggression between interspecific partners when they encountered each other (Figure 1; *t* = −8.35, *df* = 8, *p* < 0.0001).Tandem behavior showed that conspecific mating preference was indistinct between *R. chinensis* and *R. flaviceps*. Similar to tandem behavior in intraspecies, the frequencies of tandem behavior in interspecies decreased with an increase in time, which began with encountering each other and ended after one hour. The frequencies of both conspecific (Figure 2A; GLMM: *F* = 12.24, *p* < 0.001) and interspecific tandem behavior (Figure 2A; GLMM: *F* = 19.25, *p* < 0.001) were significantly different during each time of observation. However, there were no significant differences in the tandem frequencies at each time of observation between conspecific and heterospecific species in the laboratory (Figure 2A; GLMM: *F* = 0.34, *p* = 0.56). Similarly, there were no significant differences in tandem duration at each time between conspecific partners and heterospecific partners (Figure 2B; *t* = 2.31, *p* = 0.22). Our results indicated that the preferences for conspecific individuals were absent in tandem behavior.When conspecific and interspecific partners were present in the same arena, we found that there were the similar frequencies of allogrooming in early encountering between interspecies and intraspecies (Figure 3A; *t* = 0.43, *p* = 0.67). The allogrooming frequencies of both interspecies and intraspecies increased with time, whereas the allogrooming frequency of interspecies was almost twice as much as that of intraspecies (Figure 3A; GLMM: *F* = 25.85, *p* < 0.001). Significantly higher allogrooming frequency was found in interspecies as compared with interspecies (Figure 3A; GLMM: *F* = 6.39. *p* = 0.012). Although the allogrooming duration in interspecific partners was longer than intraspecific partners, no significant differences were observed between them (Figure 3B; *t* = −0.96, *p*= 0.34). These results suggested that the two species preferred interspecific versus intraspecific partners, based on allogrooming duration and frequency under laboratory conditions.Both tandem and allogrooming are associated with courtship, which allows termites to prepare for mating. We observed multiple occurrences of mating in each nest after pair formation. Mating behavior happened multiple times (average 2.34 ± 0.83 times, including mating of intraspecies and interspecies) within 1 h (see Appendix A). Strikingly, both conspecific and interspecific mating was commenced over a short time. Our result indicated that there were 56 mating behavior in period of observation, 39.29% (22/56) mating occurred in interspecies couples and 61.71% (34/56) mating occurred in intraspecies couples. However, there were no significant differences in mating frequency between conspecific and heterospecific partners (Figure 4A; GLMM: *F* = 2.53, *p* = 0.13). There were no significant differences in the mating duration between intraspecies and interspecies also (Figure 4B; *t* = 1.27, *p* = 0.21). Genotyping analyses of larvae from artificial colonies identified the number of larvae produced by hybridization and conspecific partners in artificial colonies (Appendix A). Although the accurate proportion of hybrid offspring in colonies was unknown because of limited diagnostic alleles, we still proposed that in the case of intraspecific mating being present in a colony, the interspecies mating can also produce living offspring in termites (Figure 5).

## 4. Discussion

The dynamics of hybridization among populations depends on behavioral interactions among individuals [16,17]. Our results showed that agonism between dealates was infrequent and transitory when interspecies individuals encountered each other during dispersal flight. A lack of preferences for conspecifics in the search for partners does not easily form a pre-mating barrier according to this study, and even preference for heterospecific partners is found in allogrooming behavior. Our results suggested that the pre-mating isolation mechanism may be not prevent hybridization between *R. flaviceps* and *R. chinensis*, which facilitates gene flow between the two species.

Preference for conspecifics is a type of assortative mating which is considered to be a mechanism to prevent gene exchange between distinct species or divergent populations [2,18]. Previously, assortative mating has been used to describe a particular form of mate choice in which individuals select mates on the basis of homologous genotypic or phenotypic traits [19,20,21,22]. It can contribute to the genetic isolation of two populations that come into a hybrid zone and prevent them from merging back into a hybrid population [23,24]. However, these mechanisms can be plastic, and an individual can either accept or reject a potential mate depending on the fitness of their offspring. For example, spadefoot toads, *Spea bombifrons* Cope and *Spea multiplicata* Cope in the southwestern United States preferentially mate with heterospecific partners under conditions of insufficient water, and hybrids thus gain a fitness advantage [1]. If partner resources in intraspecies are rare [25] or predators exist, choosers may favor to mate with heterospecific partners [26].

Swarming behavior of alates is important for reproductive success in termites, certainly along with high risk. During swarming, many alates leave the natal colony and randomly disperse, but less than 1% are able to found a new colony due to predators, dispersal distance, sex ratios, and other factors [27]. Thus, overzealously seeking conspecific out-breeding partners may result in the loss of an individual’s mating opportunities. For dealate termites, failing to mate means entomopathogenic infections [28], capture by natural enemies, and loss of all fitness. However, accepting a heterospecific partner permissively may allow an individual to experience higher fitness costs because of incompatible epistatic interactions between genes from different species. Although hybrid offspring may be sterile, the established heterospecific colonies many obtain high fitness that would promote genetic diversity [5,6]. In this study, no preference for conspecifics in mating behavior and genotyping strongly demonstrated that hybridization happened between the two sympatric *Reticulitermes* species under laboratory conditions when they encountered each other.

Interspecific mating increases heterozygosity, decreases inbreeding depression, and facilitates the maintenance of sexually antagonistic variation [29]. It can result in introgression from one population into another and can drive divergence during speciation with the adaptation process [30]. However, there should be mechanisms for maintaining independent species in these species when genetic material strides across the species boundary. For example, hybridization is favored in females but not in males, and hybrid offspring mating prefers one species of parents in the ants *Formica aquilonia* Yarrow and *Formica polyctena* Foerster [31,32,33]. Similar patterns might also be present in termites when hybridization occurs, but new speciation through hybridization is unusual. Other studies showed that after a closed and monogamous hybrid colony was established, colony life was not limited by food availability [34] or lifespan of founder inconformity [35,36]. Therefore, replacement and backcross can be continued in a colony if hybrid offspring are fertile, and meanwhile continuous backcross may be able to make the genetic construction of the hybrid colony come back to the original species’ gene pool. In other words, hybridization makes genetic exchange between species possible. However, lifetime asymmetry of the founder may result in backcross which can prevent gene differentiation and then eliminate the speciation.

## 5. Conclusions

Our studies showed that interspecific hybridization occurred between two sympatric *Reticulitermes* species under laboratory conditions. With interspecific hybridization, termites may increase the mating opportunities for dealates and may transfer their genes to neighbor species. The lack of preference for conspecifics in mating behavior may also benefit themselves because the offspring produced by hybridization may obtain heterosis, such as strong immunity [28] and greater escape speed from natural enemies [7,37]. We predict that hybridization between these two sympatric *Reticulitermes* species might happen in nature. This needs to be examined and proven or disproven in field observation and experimentation.

## Figures and Tables

**Figure 1 insects-11-00014-f001:**
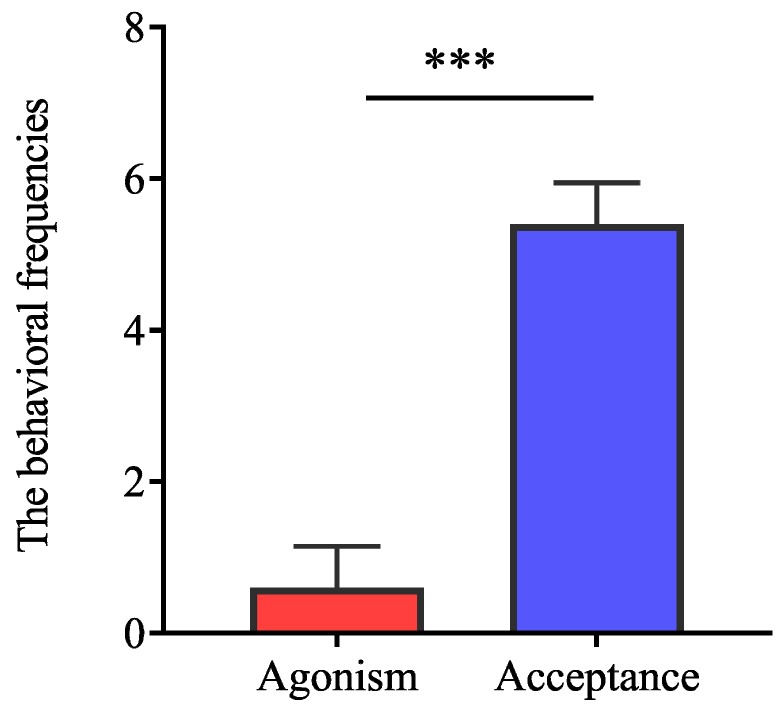
The frequencies of agonism and acceptance between the interspecific partners in the termites *R. chinensis* and *R. flaviceps*. The frequencies of acceptance are significantly higher than those of agonism between the interspecific partners (*t* = −8.35, *df* = 8, *p* < 0.0001). *** indicates *p* < 0.001.

**Figure 2 insects-11-00014-f002:**
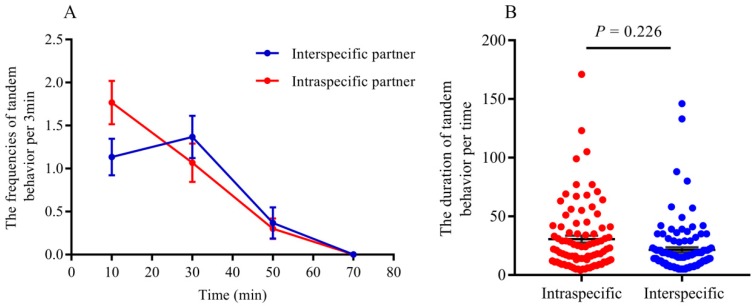
(**A**): The frequencies of tandem behavior observed in intraspecific and interspecific partners. (**B**): The duration of tandem behavior per time (unit: min). There were no significant differences in frequencies of tandem behavior between intraspecific and interspecific partner at each observing point (GLMM: *F* = 0.34, *p* = 0.56). Similarly, there were no significant differences in the duration of tandem between them (*t* = 2.31, *p* = 0.22).

**Figure 3 insects-11-00014-f003:**
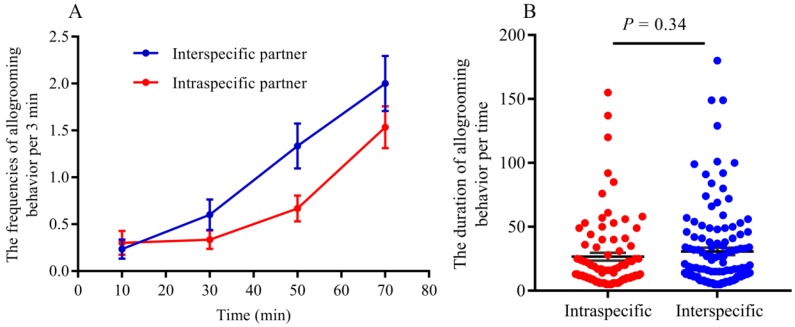
(**A**): The frequencies of allogrooming in intraspecific and interspecific partners. (**B**): The duration of allogrooming behavior per time (unit: min). The allogrooming frequencies in interspecific partners is significantly higher than those in intraspecific partners (GLMM: *F* = 6.39, *p* = 0.012), but there were no significant differences in the duration of allogrooming between them (*t* = −0.96, *p* = 0.34).

**Figure 4 insects-11-00014-f004:**
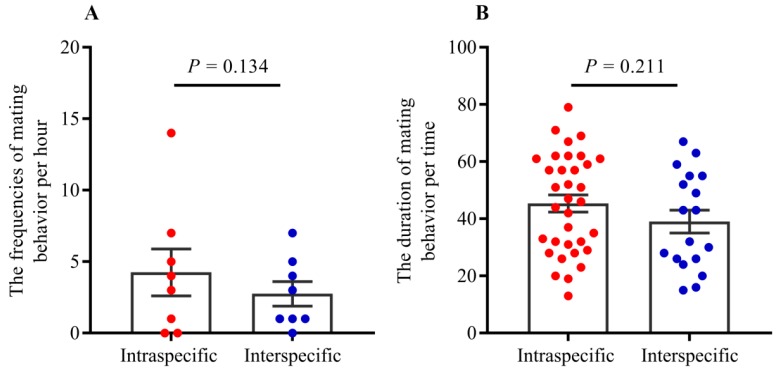
(**A**): The frequencies of mating in intraspecific and interspecific partners. (**B**): The duration of mating behavior per time (unit: min). There were no significant differences in mating frequencies per hour (GLMM: *F* = 2.53, *p* = 0.13) and durations (*t* = 1.27, *p* = 0.21) between intraspecific and interspecific partners.

**Figure 5 insects-11-00014-f005:**
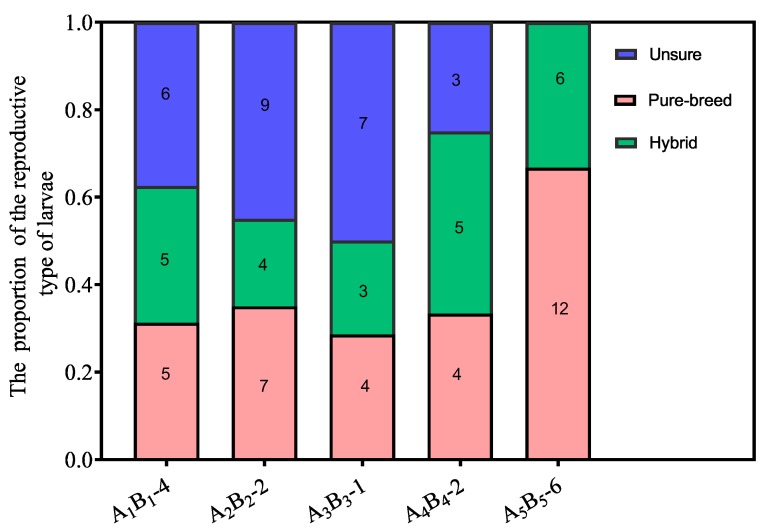
The proportion of larvae from the different mating types. The number in bar represent the number of larvae from this mating types in each colony.

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
