# Peer review of "Interspecific Hybridization between the Two Sympatric Termite Reticulitermes Species under Laboratory Conditions"

_insects, 2019, doi:10.3390/insects11010014_

Round 1

Reviewer 1 Report

Wu et al. have a very interesting study system of sympatric Reticulitermes species that are capable of hybridizing. However, the submitted manuscript is like a rough draft with many mistakes and omissions. Many details were lacking in the methods and results sections that make the study impossible to properly evaluate. For example, the methods section only mentions that the durations of each behavior were measured, but the results and figures indicate that both duration and frequency were measured. Similarly, the behavior of "acceptance" was not defined in the methods section, and yet the frequencies of this behavior were somehow measured and statistically compared to the frequencies of aggression. As another example, the number of mating pairs observed was not explained and seemed inconsistent between the results and the figures. Line 145 in the results section says 39.29% of mating was interspecific while 61.71 was intraspecific, but the figure 4A and Table S1 imply that there were exactly 8 mating events of each type. Also it is not clear how many incipient colonies were investigated for larva counts and genotyping. The table S2 only lists 4 colony types and groups all larvae together. Were there only 4 incipient colonies? Did mating occur between conspecifics and interspecifics repeatedly in each arena, or did individuals form lasting pairs? Without this information, it is difficult to see the correspondence between the mating data and the conclusions. Finally, it is a major concern that the authors conclude that the number of larvae from interspecific and intraspecific matings were not significantly different. In fact, the authors cannot know this because the proportion of undetermined larval genotypes was similar to each of the known genotype categories, and could easily contribute to a significant difference between "pure-bred" and hybrid larvae if it were biased. Much more detail about the study design and results need to be provided before this study can be properly evaluated for its scientific soundness. 

Author Response

Response Letter

Dear reviewer:

Thank you so much for positive and constructive comments to revise our manuscript (SREP-19-17519) entitled “Interspecific hybridization between the two sympatric termite Reticulitermes species under laboratory conditions”. According to your comments, we have carefully revised this manuscript. The responses to your comments and the main corrections in this paper are as follows.

Comments:

1). Many details were lacking in the methods and results sections that make the study impossible to properly evaluate. For example, the methods section only mentions that the durations of each behavior were measured, but the results and figures indicate that both duration and frequency were measured.

Response: Thanks for your comments. We described the way of measure to duration and frequency of each behavior in the methods section in this version. Please see lines 104-109.

2). Similarly, the behavior of "acceptance" was not defined in the methods section, and yet the frequencies of this behavior were somehow measured and statistically compared to the frequencies of aggression.

Response: According to your comments, we have added the defined of acceptance in the revised MS. “If there is no aggression between encountering individuals, we assume that they accept each other”. Please see lines 96-97.

3). As another example, the number of mating pairs observed was not explained and seemed inconsistent between the results and the figures. Line 145 in the results section says 39.29% of mating was interspecific while 61.71 was intraspecific, but the figure 4A and Table S1 imply that there were exactly 8 mating events of each type.

Response: Four groups were used in the behavioral observation of mating. We did 2 replicates for each group and thus there were 8 replicates. There were multiple mating events in each replicate. There were 56 mating events in period of observation. We revised these sentences in methods (see line 89) and results (see line 162) in this version.

4). Also it is not clear how many incipient colonies were investigated for larva counts and genotyping. The table S2 only lists 4 colony types and groups all larvae together. Were there only 4 incipient colonies?

Response: We used the five colonies among thirty incipient colonies in this study to analyze larva genotyping. According to your comments, we divided larvae in each colony in presenting results section. Please see Figure 5.

5). Did mating occur between conspecifics and interspecifics repeatedly in each arena, or did individuals form lasting pairs?

Response: Mating occurred between conspecifics and interspecifics repeatedly in each arena.

6). Without this information, it is difficult to see the correspondence between the mating data and the conclusions. Finally, it is a major concern that the authors conclude that the number of larvae from interspecific and intraspecific matings were not significantly different. In fact, the authors cannot know this because the proportion of undetermined larval genotypes was similar to each of the known genotype categories, and could easily contribute to a significant difference between "pure-bred" and hybrid larvae if it were biased.

Response: Thanks for your comments. We divided the types of larvae in each colony in presenting results section (Figure 5) and re-wrote this result in the revised MS. The improved sentence is “Although the accurate proportion of hybrid offspring in colonies was unknown because of limited diagnostic alleles, we still proposed that in the case of intraspecific mating present in colony, the interspecies mating can also produce living offspring in termites”. Please see 156-159.

Thanks again for your great work.

With kind regards

Sincerely yours,

Qiuying Huang

PhD and Professor

Reviewer 2 Report

Please be aware of the overuse of the word "the" when writing in English.  When in doubt leave it (the) out.

Have mostly corrected grammar and some wording.

Some sentences need clarification because you are not using the correct word to describe your finding.

Science appears to be sound.  It's a nice set of tests.

Please use species authority at first mention of genus and species in the text.  Example Coptotermes formosanus Shiraki...where Shiraki is the describing authority.  Use after genus and species at first mention only.  You do not have to use in the abstract (that however depends on the rules of this journal).

Please review comments carefully.  Some new words have been inserted and some words are inserted where words were crossed out.

Author Response

Response Letter

Dear reviewer

Thank you so much for positive and constructive comments to revise our manuscript (SREP-19-17519) entitled “Interspecific hybridization between the two sympatric termite Reticulitermes species under laboratory conditions”. According to your comments, we have carefully revised this manuscript. The responses to your comments and the main corrections in this paper are as follows.

Comments:

1) This sentence is confusing….please clarify (31-32)

Response: Thank for your suggestions. We have improved this sentence. The improved sentence is “It can be minimized or reversed depending on the fitness of offspring and environments”. Please see line 32-33.

2) Please add species authority at first mention of a species throughout the manuscript.

Response: According to your comment, we have added species authority at first mention of a species throughout the manuscript. Please see Nasutitermes corniger Motschulsky and Nasutitermes ephratae Holmgren in line 39, Coptotermes formosanus Shirak and Coptotermes gestroi Wasmann in line 40-41, Reticulitermes flaviceps Oshima and Reticulitermes chinensis Snyder in line 47 and Formica aquilonia Yarrow and Formica polyctena Foerster in line 240.

3) Why is this mention of genitalia important for termites? Please elaborate (41-42 )

Response: The genitalia are physical barriers to prevent interspecies mating. Neither queens nor kings possess external sclerotized genitalia, so interspecific mating is possible.

4) Sentence is confusing. Clarify “Even though there is a slight difference in peak swarming between species, their dispersal and reproductive periods often overlap…” (48-49)

Response: We have revised this sentence. The improved sentence is “Despite asynchronous peak flights, there is some overlap in their dispersal flight seasons”. Please see line 52-53

5) What were relative humidity/moisture conditions…..if any? (64-65)

Response: The moisture was 75% in our experiment. This information have provided in the revised manuscript. Please see line 70.

6) What was this? A dye ? lnk? Describe

Response: This is color marker (uni-Paint markers PX-21). We have described in the revised manuscript. Please see line 80.

7) What were the other dimensions of these containers? Was there a lid? Were they flat bottom or pointed or round?

Response: The containers were 120-ml transparent cylindrical vial (f = 3 cm). We have added the information of containers in the revised manuscript. Please see line 91.

8) This needs to be clarified. “cleaning of each other bodies which we define as a nuptial gift for mating….. ”(87-88)

Response: Thank for your suggestions. We have improved this sentence. The revised sentence was “We define a cleaning of each other’s bodies as a nuptial gift for mating”. Please see line 99-100.

9) You have interspecies twice here. Should the second “interspecies” be intraspecies. Check this wording. 

Response: We have replaced “interspecies” by “intraspecies”. Please see line 151.

10) this makes little sense. Please clarify and re-write this sentence. The word “vulnerable” is not the proper word needed here.(186-188)

Response: Thank for your comments. We have re-write this sentence. The re-wrote sentence was “A lack of preference for conspecifics in search for partner is difficult to form pre-mating barrier” in the revised manuscript according to this study”. Please see line 205-206.

11) Do you mean increased immunity?(233)

Response: Yes. We mean increased immunity.

Thanks again for your great work.

With kind regards

Sincerely yours,

Qiuying Huang

PhD and Professor

Reviewer 3 Report

Supplementary comments:

Table S1: italic for binomials

Table S2: l.c. for "annealing"

Table S3: footnote: replace "diagnosis" with "diagnosed"

Author Response

Response Letter

Dear reviewer

Thank you so much for positive and constructive comments to revise our manuscript (SREP-19-17519) entitled “Interspecific hybridization between the two sympatric termite Reticulitermes species under laboratory conditions”. According to your comments, we have carefully revised this manuscript. The responses to your comments and the main corrections in this paper are as follows.

Comments:

1). Replace “aggression” with “agonism” throughout paper

Response: Thank for your suggestions. We have replaced “aggression” with “agonism” throughout paper.

2). Give author names off all binomials in the first mention of sp.

Response: According to your comment, we have added species authority at first mention of a species throughout the manuscript. Please see Nasutitermes corniger Motschulsky and Nasutitermes ephratae Holmgren in line 39; Coptotermes formosanus Shirak and Coptotermes gestroi Wasmann in line 40-41; Reticulitermes flaviceps Oshima and Reticulitermes chinensis Snyder in line 47 and Formica aquilonia Yarrow and Formica polyctena Foerster in line 240.

3). Main results should not be in the introduction but in the abstract and results/discussion sections

Response: Thank for your suggestions. We have removed main results in the introduction.

4). Table S1does not give flight season or sampling data.

Response: According to your comment, we have added the information of flight season in Table S1.

5). Be specific about how the abdomens were marked

Response: The abdomens were marked by using uni-Paint markers PX-21. We have provided the information of in the revised manuscript. Please see line 80.

6). Identify and describe figures as A and B separately. What is the time unit in B?

Response: Thank for your suggestions. We have separately described figures A and B. the time unit is minute in B of figure 2, figure 3 and figure 4.

7). How old were the colonies when larvae were assayed?

Response: The larvae were assayed three months after post-colony-establishment. We have added this information in the revised manuscript. Please see line 114-115.

Thanks again for your great work.

With kind regards

Sincerely yours,

Qiuying Huang

PhD and Professor

Round 2

Reviewer 1 Report

The authors have addressed my major concerns and the manuscript is improved.